# Spin-Crossover and Slow Magnetic Relaxation Behavior in Hexachlororhenate(IV) Salts of Mn(III) Complexes [Mn(5-Hal-sal_2_323)]_2_[ReCl_6_] (Hal = Cl, Br)

**DOI:** 10.3390/ijms231911449

**Published:** 2022-09-28

**Authors:** Aleksandra V. Tiunova, Anna V. Kazakova, Denis V. Korchagin, Gennady V. Shilov, Sergey M. Aldoshin, Aleksei I. Dmitriev, Mikhail V. Zhidkov, Konstantin V. Zakharov, Eduard B. Yagubskii

**Affiliations:** 1Faculty of Fundamental Physical and Chemical Engineering, Lomonosov Moscow State University, 119991 Moscow, Russia; 2Federal Research Center of Problems of Chemical Physics and Medicinal Chemistry RAS, 142432 Chernogolovka, Russia; 3Department of Low Temperature Physics and Superconductivity, Lomonosov Moscow State University, 119991 Moscow, Russia

**Keywords:** spin-crossover, Mn^III^(sal_2_323) complexes, hexahalorhenates, polyfunctional compounds, magnetic properties, single-ion magnet

## Abstract

The cationic complexes of Mn(III) with the 5-Hal-sal_2_323 (Hal = Cl, Br) ligands and a paramagnetic doubly charged counterion [ReCl_6_]^2−^ have been synthesized: [Mn(5-Cl-sal_2_323)]_2_[ReCl_6_] (**1**) and [Mn(5-Br-sal_2_323)]_2_[ReCl_6_] (**2**). Their crystal structures and magnetic properties have been studied. These isostructural two-component ionic compounds show a thermally induced spin transition at high temperature associated with the cationic subsystem and a field-induced slow magnetic relaxation of magnetization at cryogenic temperature, associated with the anionic subsystem. The compounds are the first examples of the coexistence of spin crossover and field-induced slow magnetic relaxation in the family of known [Mn^III^(sal_2_323)] cationic complexes with various counterions.

## 1. Introduction

At present, the main class of Mn(III) spin-crossover compounds are cationic complexes of Mn(III) with the sal_2_323 ligand and its derivatives [1] (Figure 1), which were first obtained in 2003 [2], and the first spin transition in this class was discovered in 2006 [3]. Since that time, active investigations of these complexes started, aimed at studying the effects of substituents on the phenolate nucleus of the ligand, the nature of counterions and lattice solvents on the spin behavior of the complexes. The sal_2_323 ligand is a condensation product of two molecules of salicylaldehyde with a flexible tetraamine 1,2-bis(3-aminopropyl-amino)ethane, Figure 1. The use of this tetramine leads to a trans-arrangement of the oxygen atoms of the sal_2_323 ligand around the manganese ion. As a result, the octahedral configuration of Mn(III) is strongly distorted and axial contraction of the Mn-O bonds occurs. Such a molecular structure of Mn(III) complexes with ligands of the sal_2_323 family contributes to the appearance of reversible spin transitions from the S = 1 state to the S = 2 state upon heating (spin-crossover (SCO) behavior), accompanied by the population of d_x_^2^_-y_^2^ orbital. During the transition, the equatorial Mn-N bonds lengthen, while the axial Mn-O bonds remain practically unchanged.

To date, several important results have been obtained in this class of Mn(III) spin-crossover complexes. Variation of substituents and counterions led to the synthesis of the complexes, which showed sharp spin transitions with significant hysteresis [4,5,6,7,8,9,10,11,12,13], including a conducting spin-crossover complex with an electroactive counterion TCNQ with a hysteresis width of 50 K, [Mn(5-Cl-sal_2_323)]TCNQ_1.5_·2CH_3_CN [7]. The Mn(III) complexes with a 3,5-dihalo-substituted sal_2_323 ligand and tetraphenylborare as a counterion were recently synthesized and showed unusual structural and magnetic properties [8,9,10,11,12]. So, the dichlorine-containing complex displayed three structural phase transitions below 300 K and two spin crossovers coupling with structural transitions. One of the spin crossovers is characterized by the formation of an intermediate HS:LS phase from the initial high-spin state (HS), and the second is associated with the transition from the HS:LS phase to the low-spin state (LS) [11,12]. The giant magnetoelectric coupling was found at a Mn(III) spin crossover in [Mn(3,5-diBr-sal_2_323]BPh_4_ [14].

In this work, we used for the first time a doubly charged paramagnetic anion [ReCl_6_]^2−^ as a counterion in the Mn(III) cationic complexes with a monohalo-substituted ligand (5-Cl(Br)-sal_2_323). To the best of our knowledge, all known complexes of the [Mn(sal_2_323]^+^ family synthesized so far contain singly charged anions. An important argument in favor of choosing the [ReCl_6_]^2−^ as a counterion was the fact that this anion demonstrates a field-induced single-ion magnet behavior [15], like its analogs [ReBr_6_]^2−^, [ReI_6_]^2−^ and [ReF_6_]^2−^ [16,17,18]. Thus, one could expect the formation of complexes combining the Mn(III) spin crossover and [ReCl_6_]^2−^ single-ion magnetism in the same crystal lattice. Only a few compounds are known in the literature in which the spin transition coexists with single-molecule magnetism [19,20,21,22,23,24,25,26,27]. Two isostructural complexes [Mn(5-Cl-sal_2_323)]_2_[ReCl_6_] (**1**) and [Mn(5-Br-sal_2_323)]_2_[ReCl_6_] (**2**) were synthesized. Their crystal structures and magnetic properties have been studied. Herein, we present these results.

## 2. Results and Discussion

### 2.1. Synthesis

Complexes **1** and **2** were synthesized according to Figure 1 in a solvent containing 96% ethanol and acetonitrile in a ratio of 1:1, see Section 3. Previously, the 5-Cl-sal_2_323 and 5-Br-sal_2_323 ligands were synthesized using the N,N′-bis(3-3-aminopropyl)ethylenediamine and the corresponding salicylaldehyde. Then, a solution containing Mn(NO_3_)_2_ 4H_2_O and K_2_[ReCl_6_] was added to the resulting solution of ligands. After mixing the two solutions, the color changed from yellow to dark red, which indicates the oxidation of Mn(II) to Mn(III). Black single crystals **1** and **2** were formed upon slow evaporation of the solvent.

### 2.2. Crystal Structures

The compounds **1** and **2** crystalize in the monoclinic system (centrosymmetric space group *C2/c*) and are isostructural. Both compounds do not undergo a temperature-induced phase transition in the studied range from 100 to 423 K. Electroneutral unit of **1** and **2** consists of two mononuclear cationic [Mn(5-Cl(or Br)-sal_2_323)]^+^ complexes and one [ReCl_6_]^2−^ counter anion (Figure 1). Asymmetric units are represented by halves of these moieties, the Mn and Re ions are on special positions. ReCl_6_^2−^ anions have a slightly distorted octahedral environment with bond lengths in a narrow range (2.3511(6)–2.3765(6) Å for **1** and 2.3505(8)–2.3760(7) Å for **2**). These values are in agreement with those in known compounds with ReCl_6_^2−^ anions [15,28].

The [Mn(5-Cl(or Br)-sal_2_323)]^+^ complexes have a distorted hexacoordinated geometry with MnN_4_O_2_ coordination environment corresponding to the previously reported structures of cationic complexes [Mn(R-sal_2_323)]^+^ with similar hexadentate ligands [1]. The bond lengths Mn-N(or O) and octahedral distortion parameters (ζ, Σ and Θ) in coordination polyhedron Mn^III^N_4_O_2_ of **1** and **2** (for different temperatures) are shown in Table 1. Two nonequivalent [Mn(5-Cl(or Br)-sal_2_323)]^+^ cations found in the structures of the compounds under consideration have significantly different Mn-N bond lengths and octahedral distortion parameters.

At 100 and 240 K, these nonequivalent Mn ions correspond to different spin states (LS and HS for Mn(1) and Mn(2), respectively). The average equatorial bond lengths Mn–N_im_ and Mn–N_am_ in complexes **1** and **2** fall in the range of that expected for LS and HS Mn(III) ions (Table 1) [1]. Upon increasing the temperature to 300K, the bond lengths Mn(1)–N show a slight increase in values that corresponds to the part of low-spin Mn(1) ions transit to a high-spin state (Table 1). A further gradual increase in temperature to 423 K reveals an increase in the bond lengths Mn(1)–N_im_ and Mn(1)–N_am_, which corresponds to an increase in the fraction of HS Mn(1) ions in the crystals, while these values do not reach values corresponding to the HS Mn(III) ion. For second nonequivalent ion Mn(2), the bond lengths and angles in coordination polyhedron correspond to the Mn(III) ion in HS state (Table 1) and are retained in all temperature ranges under consideration. The axial Mn–O bond lengths do not change significantly with increasing of temperature for both nonequivalent Mn ions.

Besides the bond lengths changing, strong changes in the bond angles X-Mn-X are observed, which is well illustrated by the significant difference in the parameters of octahedral distortion (Σ and Θ) shown in Table 1. For complexes **1** and **2**, the octahedral distortion parameters for Mn(1) complexes at 100 and 240 K are close and corresponding to the Mn(III) ion in LS state [1]. Upon increasing the temperature from 240 to 423 K, the values of the octahedral distortion parameters for Mn(1) increase from ~45–46 to ~69 and from ~129–133 to ~193–194° for Σ and Θ, respectively. At 300K and higher, the octahedral distortion parameters for Mn(1) ions have intermediate values between ones for LS and HS Mn(III) ions, that may indicate on an incomplete gradual spin transition between LS (S = 1) and HS (S = 2) states of Mn(III) ion in complexes under consideration. At the same time, Mn(2) coordination polyhedra have the octahedral distortion parameters (Table 1) corresponding to the high-spin Mn(III) ion, which are not changed significantly with increasing of temperature from 100 to 423 K.

The crystal structures of **1** and **2**, besides electrostatic cation–anion interactions, are stabilized by intermolecular H-bonds and weak C-H…π_Ph_ and C-H…Hal intermolecular interactions. The key intermolecular interactions are non-covalent N–H⋯Hal interactions between [Mn(5-Hal-sal_2_323)]^+^ cations. Only Mn1 complexes that undergo a high-temperature spin transition are linked into chains with each other due to intermolecular N-H…Cl(Br) hydrogen bonds along *c*-axes (Figure 2). The geometric parameters of these H-bonds are shown in Table 2.

The Mn2 complexes, which remain high-spin over the entire temperature range studied, weakly bound with neighboring the Mn2 complexes and ReCl_6_^2−^ anions due to C-H…Cl(Br) contacts 2.77 (2.80 for **2**) Å and have also discrete intermolecular contacts C-H…π_Ph_ 2.67 (2.67 for **2**) Å with the Mn1 complexes. Mn1 complexes have the same C-H…Cl (ReCl_6_) contacts 2.78 (2.80 for **2**) Å, which apparently do not influence the spin transition process in these salts. Thus, strong non-covalent intermolecular interactions between cations and anions that can play a significant role in the spin-state switching behavior are absent in crystal packing of **1** and **2** [1,6,7,8,9,10,11,12,29,30]. As a result, we can observe only an incomplete high-temperature spin transition in Mn1 complexes that are linked into chain by N-H…Hal intermolecular H-bonds. The Mn and Re centers are well isolated from each other in the crystal structure, with the shortest intermolecular Mn...Mn, Mn...Re, and Re...Re distances being equal to 7.85(7.87 for **2**), 7.33(7.34), and 11.97(12.03) Å, respectively.

### 2.3. DC Magnetic Properties

Static magnetic susceptibility was measured on samples of complexes **1** and **2** in a field of 0.5 T at temperatures of 2–400 K. The phase purities of these bulk samples were confirmed by powder X-ray diffraction (Appendix A). Both compounds exhibit a very similar behavior (Figure 3). The *χ_m_T* vs. *T* dependencies are close, and the curves remain the same upon heating and cooling. At temperature below 240 K, the *χ_m_T* value slowly decreases, reaching 5.12 cm^3^K/mol and 5.07 cm^3^K/mol for **1** and **2**, respectively, at 100 K. These values are close to the calculated value (5.6 cm^3^K/mol) for the sum of three magnetically non-interacting centers: Re(IV) ion with *χ_m_T* = 1.6 cm^3^K/mol (S = 3/2 with g = 1.80–1.90) [31,32] and two independent Mn (III) ions: Mn2 in high-spin (HS) state with *χ_m_T* = 3.0 cm^3^K/mol (S = 2, g = 2), and Mn1 in the low-spin (LS) state with *χ_m_T* = 1.0 cm^3^K/mol (S = 1, g = 2). There are good correlations between these magnetic data and the Mn-N bonds lengths at 100 K, which clearly indicate that the cations in complexes **1** and **2** are in different spin states: Mn1 is in the LS state, while Mn2 is in the HS state. With an increase in temperature from 240 K, a gradual increase in the *χ_m_T* value is observed, which reaches values of 6.67 cm^3^K/mol and 6.96 cm^3^K/mol at 400 K for complexes **1** and **2**, respectively, indicating a gradual transition of the Mn1 cations in both complexes from the LS→HS states. However, by 400 K, complete spin transformation does not occur. An analysis of the crystal structures of **1** and **2** at temperatures above 400 K (423 K) showed that the Mn-N bond lengths continue to increase with a further increase in temperature. Thus, in both complexes at high temperatures, the Mn(III) cationic subsystem is in the high-spin state, while at low temperatures (below 240), an intermediate state is realized, in which half of the cations are in the high-spin state (Mn2) and the other (Mn1) in a low spin state.

Upon cooling below 100 K, the *χ_m_T* values for these compounds continue to slowly decrease at first, and then, around 40-30 K, they decrease abruptly to reach 1.99 (**1**) and 1.80 cm^3^K/mol (**2**) at 2 K. The abrupt decrease in the *χ_m_T* values is mainly due to the strong magnetic anisotropy, which are observed in Re^IV^- and [Mn^III^(sal_2_323)]-based systems [31,32,33,34,35]. These *χ_m_T* values at 2.0 K are comparable to the sum of the *χ_m_T* values for the isolated Re(IV) complexes with halide ligands (approx. 1.0 cm^3^K/mol) [15,16,32] and the Mn(III) complexes in mixed HS:LS = 1:1 state with the sal_2_323 ligand family (approx. 0.7–0.9 cm^3^K/ mol) [9,11,36] at low temperature. Such a correspondence indicates that the antiferromagnetic coupling between the complexes is very weak in compounds **1** and **2**, which correlates with the strong isolation of Mn and Re centers in the crystal structure. It should be noted that the observed slow linear decrease in the *χ_m_T* values upon cooling from 240 to 30 K is possibly associated with the contribution of the temperature-independent paramagnetism (TIP).

The field dependences of the magnetization (*M* vs. *µ*_0_*H/T*, Figure 4) for complexes **1** and **2** were recorded at temperatures 2, 3, and 5 K in the field range of 0–5 T. For both complexes, the magnetization reaches a value of about 4.0 at 5.0 T without saturation. The low value of magnetization at 5.0 T, absence of saturation and the non-coincidence of M/μ_B_ vs. *µ*_0_*H/T* curves at different temperatures indicate the presence of considerable magnetic anisotropy in the complexes as a results to significant zero-field splitting effects inherent in the Re(IV) and Mn(III) ions [15,16,17,31,32,33,34,35,37,38,39].

### 2.4. AC Magnetic Properties

In order to test whether the complexes **1** and **2** exhibit the properties of single-ion magnet (SIM), the ac susceptibility of both compounds was studied in the absence and at applied static magnetic field. The frequency dependences of the in-phase *χ*′(ν) and non-zero out-of-phase *χ*″(ν) signals of the ac susceptibility make it possible to test the relaxation mechanisms in magnetic systems [40,41]. Both compounds do not display the slow relaxation of the magnetization in the absence of the applied magnetic field; the *χ*″ signals are not observed. It is known that magnetic relaxation of SMMs is highly susceptible to quantum tunneling of magnetization (QTM), which can be suppressed by an external applied dc field. The field-dependencies of ac susceptibility (*χ*′ and *χ*″) were recorded at 2 K in fields of 0–3000 Oe and two fixed frequencies (200 and 1000 Hz) in search for an optimal field to suppress the QTM effect. The value of the optimal magnetic field corresponds to the position of the maximum in the dependence of *χ*″(H). With an increase in the field, the *χ*″ increases and reaches some saturation at 3000 Oe (Appendix A). To test the relaxation behavior of compounds **1** and **2**, the ac susceptibility was studied in a field *H_dc_* = 3000 Oe at frequencies of 100–10,000 Hz and temperatures of 2–3 K (Figure 5 and Appendix A). Frequency-dependent signals from both components of the ac susceptibility were observed at temperatures from 2.0 to 2.75 K, indicating field-induced slow relaxation of magnetization and, therefore, SIM phenomenon. However, the *χ*″ maxima lie above 10,000 Hz that is not feasible in our experimental set-up. Assuming that the relaxation of magnetization in **1** and **2** is Debye process driven by the thermal activation over the energy barrier *U_eff_*, we estimated the values of energy barrier *U**_eff_* and *τ*_0_, using the formula (1) [37,42].
(1)ln(χ″/χ′)=ln(ωτ0)+UeffkT,
where *ω* = 2πν–angular frequency, *τ*_0_—preexponential factor in the Arrhenius law *τ* = *τ*_0_exp(*U**_eff_*/*kT*), *T*—absolute temperature, *τ*—time relaxation, *U_eff_*—the effective spin-reversal barrier, *k*—Boltzmann constant.

The experimental plots ln(*χ*″/*χ*′) vs. *T*^−1^ at different frequencies, their fitting by the formula (1) and values of parameters are presented in Figure 6. The values of the *τ*_0_ parameter for **1** and **2** and other similar Re^IV^ complexes are close [16,17,38]. The average values of *U**_eff_* are 13(1) and 16(1) K for **1** and **2**, respectively. The *U**_eff_* values for **1** and **2** are similar between them, and close to that of compound (Th_2_imH^+^)_2_[ReCl_6_] with diamagnetic photochromic cation Th_2_imH^+^ (*U_eff_* = 12.21 before irradiation and 11.35 K after irradiation) [15]. At the same time, these values are higher than in the case of *U**_eff_* for (Ph_4_P)_2_[ReI_6_] and approximately two and three times lower than those for field-induced single-ion magnets (Ph_4_P)_2_[ReX_6_], X = F, Br, respectively [16,17]. Research on the hexahalorhenates [Re^IV^X_6_]^2−^ (X = F, Cl, Br and I) has shown that the magnetic behavior of these anions depend greatly on the nature of the counterion [43]. For example, a study of the [ReI_6_]^2−^ ion with alkali metal cations, from Li^+^ to Cs^+^, has illustrated that none of these compounds is a SIM [39], same as in the case of hexahalorhenates with paramagnetic cations of ferrocenium [28] and metal oxazolidine nitroxides [43], whereas substitution of the perchlorate counterions in the Mn_6_ SMM by [Re^IV^Cl_6_]^2−^ led to the energy barrier for magnetization relaxation increasing by 30% [44].

## 3. Materials and Methods

The starting compounds and solvents were obtained from commercial sources and were used without further purification. The reactions related to the synthesis of the complexes were carried out in air. IR spectra were recorded on solid samples using a Bruker ALPHA spectrometer with the attenuated total reflectance (ATR) module. Elemental analyses were performed on a Vario MICRO cube (Elementar Analysensysteme GmbH, Langenselbold, Germany) equipment. Thermogravimetric analysis was carried out in a helium flow with a purity of 99.9999% using a STA 449 F3 Jupiter thermal analyzer. The heating rate was 2.0 °C/min.

### 3.1. Synthesis of [Mn(5-Cl-sal_2_323)]_2_[ReCl_6_] (1)

5-Cl-salicylicaldehyde (0.5 mmol, 0.0783 g) was dissolved in acetonitrile-ethanol mixture (1:1, 15 mL) and added with vigorous stirring to N,N′-bis(aminopropyl)ethylenediamine (0.25 mmol, 0.0457 mL), previously dissolved in the same mixture of solvents (15 mL). After mixing the solutions, stirring was continued for 15 min. Manganese(II) nitrate tetrahydrate (0.25 mmol, 0.0627 g) and potassium hexachlororenate(IV) (0.125 mmol, 0.0596 g) were dissolved in the same solvent mixture (40 mL) with 5 mL distilled water added. After adding this solution to the ligand solution, the color changed from yellow to dark red. The solution continued to be stirred intensively for another 30 min and filtered out. Black single crystals precipitated from the solution within a week during the slow evaporation of the solvents. The crystals were filtered off and dried in air. Yield: 47%. Elemental analysis, calc. (%) for C_44_H_52_Cl_10_Mn_2_N_8_O_4_Re: C, 37.54, H, 3.72, N, 7.96; found: C, 36.96, H, 3.71, N, 7.85. IR (only intense bands, cm^−1^): 3263, 2928, 2876, 1613, 1378, 1279, 1069, 985, 805, 704, 469 (Appendix A).

### 3.2. Synthesis of [Mn(5-Br-sal_2_323)]_2_[ReCl_6_] (2)

The procedure for the synthesis of complex **2** is similar to the synthesis of complex **1**, except that 5-bromosalicylaldehyde was used instead of 5-chlorosalicylaldehyde. Yield: 52%. Elemental analysis, calc. (%) for C_44_H_52_Br_2_Cl_6_Mn_2_N_8_O_4_Re: C, 33.33, H, 3.30, N, 7.07; found: C, 32.91, H, 3.32, N, 6.98. IR (only intense bands, cm^−1^): 3249, 2928, 2871, 1613, 1374, 1279, 1071, 985, 828, 682, 461 (Appendix A).

The powder XRD analysis showed the phase purity of bulk samples of complexes **1** and **2** (Appendix A). Thermogravimetric analysis of both complexes showed that they are stable up to 220 °C (Appendix A).

### 3.3. Magnetic Measurements

Dc magnetic properties of powdered samples **1** and **2** were measured by a vibrating-sample magnetometer of a Cryogen Free Measurement System (Cryogenic Ltd., UK). The dependencies *M*(*T*) were measured in cooling–heating conditions at *T* = 2.0–400 K in static magnetic field *H* = 0.5 T. The temperature change rate was 2 K/min. The magnetic moment of the samples took into account the contribution of the sample holder. Molar magnetic susceptibility *χ* was corrected taking into account a diamagnetic component of susceptibility according to the Pascal rule. Ac measurements carried out in a 3 Oe oscillating field on a magnetometer PPMS-9 (Quantum Design) in the absence and application of dc magnetic field.

### 3.4. X-ray Diffraction

X-ray diffraction data of **1** and **2** were collected at different temperatures (from 100 to 423 K) using an *Agilent XCalibur* diffractometer employing graphite monochromatic *MoK_α_* radiation (*λ*= 0.71073 Å). Experimental absorption corrections (SCALE3ABSPACK) were applied using the *CrysAlisPro* program [45] to all data. The structures were solved by direct methods and refined on F^2^ by anisotropic full-matrix least-squares methods using *SHELXTL* program [46,47]. All non-hydrogen atoms were refined in the anisotropic approximation against F^2^ for all reflections. Hydrogen-atom positions were obtained from difference Fourier syntheses and were refined with riding model constraints on their respective atoms. A summary of the crystallographic details of the X-ray analyses is given in Appendix A. Selected bond lengths and calculated using *OctaDist* software [48] octahedral distortion parameters are given in Table 1. The X-ray crystal structures data were deposited with the Cambridge Crystallographic Data Center (CCDC), with CCDC 2182503-2182508 and 2182509-2182514 reference codes for **1** and **2**, respectively. Powder X-ray diffraction patterns for **1** and **2** were registered on an Aeris diffractometer (Malvern PANalytical B.V., Netherlands) at room temperature. The comparison of powder XRD and single crystal data showed that polycrystalline samples **1** and **2** are monophase crystalline materials (Appendix A).

## 4. Conclusions

Multifunctional compounds based on spin-crossover complexes of the [Mn^III^(sal_2_323)]^+^ family with the magnetic anion [Re^IV^Cl_6_]^2−^: [Mn(5-Cl-sal_2_323)]_2_[ReCl_6_] (**1**) and [Mn(5-Br-sal_2_323)]_2_[ReCl_6_] (**2**) were synthesized. Complexes **1** and **2** are isostructural and contain in elemental unit two independent cations Mn1 and Mn2. The study of the crystal structures of the complexes at six temperatures (100, 240, 300, 350, 403, and 423 K) and their dc magnetic properties showed that cation Mn1 at 100 K is in the low-spin state with S = 1, and as the temperature rises to 423 K gradually passes into a high-spin state (S = 2), which is accompanied by an increase in the Mn-N bond lengths and a significant distortion of the Mn1N_4_O_2_ octahedral parameters (Table 1). In contrast to Mn1, cation Mn2 is in the HS state in the studied temperature range. Both components of ac susceptibility (*χ*′ and *χ*″) exhibit frequency dependent signals when a constant magnetic field is applied. This behavior of the ac susceptibility indicates slow magnetic relaxation, meaning that complexes **1** and **2** are field-induced single-ion magnets. Until now, only one case of the manifestation of single-ion magnetic behavior of the [Re^IV^Cl_6_]^2−^ anion has been known in the literature [15]. Compounds **1** and **2** are the first examples of the coexistence of a spin crossover at high temperatures and slow magnetic relaxation at low helium temperatures into the family of [Mn^III^(sal_2_323)] cationic complexes. This result opens up prospects for creation of new multifunctional compounds combining spin transition and single molecular magnetism based on cationic complexes of the [Mn^III^(sal_2_323)]^+^ with counterions–anionic single molecule magnets, proceeding from mononuclear complexes of Re(IV), Co(II), and Dy(III) with increased temperatures of magnetization barriers.

## Data Availability

Not applicable.

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
