# Peer review of "Spin-Crossover and Slow Magnetic Relaxation Behavior in Hexachlororhenate(IV) Salts of Mn(III) Complexes [Mn(5-Hal-sal2323)]2[ReCl6] (Hal = Cl, Br)"

_ijms, 2022, doi:10.3390/ijms231911449_

Round 1
Reviewer 1 Report
This article presents an interesting spin crossover and single ion magnet combination in a novel synthesized material.
Scientific point which i would like to see justified is the fact that tau_0 and U_eff are not constant but instead different values are used for every ac frequency. Please cite support for doing this and, more importantly, the physical significance. Perhaps showing best fits for each compound given single values of tau0 and U_eff can give an idea of how well the Debye model works, and deviations can be discussed more transparently.
Minor edits needed:
1) line 25: remove "an"
2) line 32: define SCO
3) line 114: ..."that may indicate an incomplete gradual spin..."
4) 137: remove "on"
5) 140: "observe an incomplete..."
6) 188: remove "an"
7) figure 4: x-axis units should be "mu_0*H/T" if you will quote H in units of tesla
8) 195: define SIM here instead of line 211
9) 198: "revealed in..."
10) 199: remove "The"
11) figure 5: x-axis should be nu instead of f for frequency, to be consistent with equations in text
12) 226: clarify here why two values of Ueff are given for one compound
Author Response
Answer: Straight lines in fig. 6 are not quite parallel, which is due to the low accuracy of determining the values of χ' and χ”, which have low values for these complexes. The main reason for the scatter in the values of t0 and Ueff at different frequencies is associated with the low accuracy of this method for determining these parameters [42]. The authors of the method claim that "a more precise result must wait for very low-temperature measurements T<1 K" [42]. Nevertheless, the values of t0 and Ueff are close at different frequencies. We are discussed in text the average values for Ueff, but these different values were also presented at different frequencies to demonstrate the accuracy of the data obtained and the approach used in their processing.
Minor edits needed:
1) line 25: remove "an"
2) line 32: define SCO
3) line 114: ..."that may indicate an incomplete gradual spin..."
4) 137: remove "on"
5) 140: "observe an incomplete..."
6) 188: remove "an"
7) figure 4: x-axis units should be "mu_0*H/T" if you will quote H in units of tesla
8) 195: define SIM here instead of line 211
9) 198: "revealed in..."
10) 199: remove "The"
11) figure 5: x-axis should be nu instead of f for frequency, to be consistent with equations in text
12) 226: clarify here why two values of Ueff are given for one compound
Answer: The compound (Th2imH+)2[ReCl6] contains a photoactive cation Th2imH+ and shows different values of the magnetization barrier before irradiation and after irradiation when applying a dc field [15]. The sentence in main text of manuscript has been corrected. All minor edits have been corrected. Figures 4 and 5 have been updated.
Reviewer 2 Report
An interesting and high-quality manuscript. The results of the study are of interest to specialists in the field of solid state chemistry and magnetochemistry.
Author Response
Thank you so much!
Reviewer 3 Report
1. The authors mentioned that the in-phase susceptibility X' is recorded. However, the data is not included in the main text, or in the supplementary information.
2. In the discussion of temperature-dependent XmT, the temperature region is split into three parts. The authors claim that the XmT decreases slowly from 100K to ~30K, and tend to be constant from 240K to 100K. However, in Figure 3 XmT shows a clear decrease vs T in the range from 240K to ~ 30K. Why the interpretation is different?
3. The authors mention that they measure the susceptibility under the field of 0-3000 Oe to find the optimal field to suppress the QTM effect. As shown in Figures S2 and S3, the field strength is not large enough to show the optimal field.
4. There is some formatting error in Figure S6.
Author Response
- The authors mentioned that the in-phase susceptibility X'is recorded. However, the data is not included in the main text, or in the supplementary information.
Answer: in-phase susceptibilities for both complexes have been included in the supplementary information as FigS7.
- In the discussion of temperature-dependent XmT, the temperature region is split into three parts. The authors claim that the XmT decreases slowly from 100K to ~30K, and tend to be constant from 240K to 100K. However, in Figure 3 XmT shows a clear decrease vs T in the range from 240K to ~ 30K. Why the interpretation is different?
Answer: We have changed corresponding parts of the text of the manuscript.
Upon cooling below 100 K, the χmT values for these compounds decrease first slowly and then, around 40-30 К, they decrease abruptly to reach 1.99 (1) and 1.80 cm3K/mol (2) at 2 K. Upon cooling below 100 K, the χmT values for these compounds continue to slowly decrease at first, and then, around 40-30 K, they decrease abruptly to reach 1.99 (1) and 1.80 cm3K/mol (2) at 2 K.
…
It should be noted that the observed slow linear decrease in the χmT values upon cooling from 240 to 30 K is possibly associated with the contribution of the temperature-independent paramagnetism (TIP).
- The authors mention that they measure the susceptibility under the field of 0-3000 Oe to find the optimal field to suppress the QTM effect. As shown in Figures S2 and S3, the field strength is not large enough to show the optimal field.
Answer: We have corrected the corresponding sentence in main text of the manuscript.
A broad maximum in the range of 3000 Oe was observed in both complexes (Figures S2 and S3). With an increase in the field, the χ” increases and reaches some saturation at 3000 Oe (Figures S2 and S3).
- There is some formatting error in Figure S6.
Answer: We do not have a complete understanding of what needs to be corrected in Fig.S6. There is the same Fig.S5 which is not asked to be corrected.